# Mathematical modeling of Dengue virus serotypes propagation in Mexico

**Gilberto Sánchez-González, Renaud Condé**  *

Centro de Investigación Sobre Enfermedades Infecciosas, Instituto Nacional de Salud Pública, Morelos, México

* renaud.conde@insp.mx

## Abstract

The Dengue virus (DENV) constitutes a major vector borne virus disease worldwide. Prediction of the DENV spread dynamics, prevalence and infection rates are crucial elements to guide the public health services effort towards meaningful actions. The existence of four DENV serotypes further complicates the virus proliferation forecast. The different serotypes have varying clinical impacts, and the symptomatology of the infection is dependent on the infection history of the patient. Therefore, changes in the prevalent DENV serotype found in one location have a profound impact on the regional public health. The prediction of the spread and intensity of infection of the individual DENV serotypes in specific locations would allow the authorities to plan local pesticide spray to control the vector as well as the purchase of specific antibody therapy. Here we used a mathematical model to predict serotype-specific DENV prevalence and overall case burden in Mexico.

## Introduction

Recent increase in Dengue virus (DENV) spread in several region of America and Asia have sparked renewed concerns about this disease. This virus is transmitted by the *Aedes* genus mosquitoes, which are currently distributed in the tropical and subtropical regions [1]. In recent years, the effects of the global warming have allowed the vector to spread northern and southern of its customary location, now being present in the south of Europe [2] and the United States [3]. Since these human populations are Dengue naive, outbreaks outcomes could result pandemic [4]. The Dengue distribution is ruled by multifactorial conditions, such as mosquito vector conditions, specific biological characteristics of the virus strain, previous immunity of the human population and the vector mosquitoes/human densities. The vector (*Aedes* mosquito) population density is influenced by the geographical height, temperature, and humidity of the studied area [5]. The virus geographic dispersion results essentially from the human displacements, either from tourism, migration (legal and illegal) or labor related travels [2].

The south-southeast regions of Mexico are endemic for dengue, showing incidence variations depending on the specific weather condition. The magnitude of the epidemic outbreaks also depends on the circulating serotypes. As a consequence, a dengue epidemiological surveillance of the 32 states of the Mexican Republic is performed by the General Directorate of

**Data Availability Statement:** Monthly meteorological data of temperature and precipitation were downloaded using the Mathematica's package "WeatherData", which

gives current and historical weather data for all available weather stations. The DENV serotype infection incidence were extracted from Falcón-Lezama J, Sánchez-Burgos GG, Ramos-Castañeda J. Genética de las poblaciones virales y transmisión del dengue. Salud Publica Mex (2009) 51: doi:10. 1590/S0036-36342009000900006.

**Funding:** The author(s) received no specific funding for this work.

**Competing interests:** The authors have declared that no competing interests exist.

Epidemiology (DGE) [6]. Even though prevention and vector control efforts have been ramped up, the Dengue cases have been multiplied by ten during the last 20 years, reaching 264,898 cases in 2019 [6]. It is worth mentioning that it has been stated that underreporting and misdiagnosis of the DENV cases limit the ability to determine the true burden of disease [7].

In Mexico, the recent occurrences of DENV 3 and 4 epidemics reported by the INDRE triggered fears of hemorrhagic Dengue outbreaks. Furthermore, the hyper-endemicity observed in distinct Mexico states has been related to an increase in disease severity [8]. The Dengue vaccine available so far in Mexico display protection between 35% and 72%, depending on the serotype [9] and present an increased risk of severe dengue in seronegative subjects [10]. Therefore, the most commonly available mitigation strategies are the vector control and the prevention of extensive exposure of the population to potential Dengue vectors (through the use of bed nets, fumigation and repellents) [11]. Since the different DENV serotypes show varying proportion of hemorrhagic outcome [12, 13] and the transition of one serotype prevalence to another usually correlates with DENV infection outburst [13], a mathematical model that would allow the prediction of individual DENV serotypes epidemic would be a useful tool for healthcare policy makers. Also, since several publication pointed that this syndrome is linked to re-infection of human showing previous immunity to the same DENV serotype [14, 15], such model could allow the prediction of the rate and proportions of hemorrhagic DENV infection.

In the mid-1970s, Dietz [16] proposed a compartmental model of Dengue diffusion that unifies the assumptions of previous models and proposes vector control and vaccination strategies applicable to arboviruses in general. It became the first model used to characterize the ecological peculiarities of DENV transmission. Such was its importance that practically all subsequent models were based on this methodology [17]. Nevertheless, this line of research was neglected until Focks et al. demonstrated the qualitative value of the models in a series of works aimed at evaluating and implementing vector control strategies. They used entomological data and simulation of outbreaks in human and mosquito populations to generate a useful tool that helps understand the epidemiology of Dengue [18, 19]. Since then, a growing number of models have analyzed particular aspects of the diseases. As a result, in 2011, the WHO Vaccine Research Initiative considered that the impact of future vaccines against DENV should be estimated using mathematical models [20].

Andraud et al. presented a detailed literature review of 42 different Dengue spread models published until the year 2011 [21]. They associate these models in function of their underlying assumptions based on epidemiological and entomological studies and explore the potential impacts of these parameters on the vector control or vaccination strategies. These models were classified in function of the number of serotypes they represented and if they considered the vector population. Eighteen of these models show the host-vector dynamic using a single serotype. The other 24 models describe a scenario of multiple serotypes with both direct (host-host) and indirect (host-vector) transmissions. Of these 23 models, three introduce seasonality, though solely as a mathematical-periodical function of time [22–24]. Recent studies consider real-time meteorological data in their estimations but these models need more development to incorporate serotype dependency [25, 26].

We previously developed a pan-serotypic mathematical model for Dengue dispersal in function of the meteorological characteristics of the chosen location [27]. Here, we further expand the model in order to take into account the particularity of the 4 serotypes, as well as the cross-immunity resulting from previous infection with one serotype or the other. This new model predicts the serotype specific DENV dynamics using the real-time meteorological data, like temperature and precipitation of a determined geographical region. To achieve prediction capacity, the

infection parameters of the model are temperature dependent. Since the temperature is a parameter of major importance, we distinguish the outdoor temperature, which drives the speed of development of the mosquito during its aquatic phase of life, from the indoor temperature, which drives the life expectancy of adult mosquitos as well as the intrinsic DENV incubation period. Additionally, this model would allow for prediction of hemorrhagic DENV infection.

## Methods

### Smoothing of the meteorological data

Monthly meteorological data of temperature and precipitation were downloaded using the Mathematica's package "WeatherData", which gives current and historical weather data for all available weather stations. As a first approach, we fed the model with the original discrete values of rainfall and temperature, but we noted the appearance of singularities due to the lack of continuity in those parameters, thus we established the following methodology for the data smoothing:

### Temperature

The data were ordered as time series (t). For each year, the minimal and the maximal values of temperature are obtained. A polynomial function $P_1$ is fitted to the minimal values of the data, and a polynomial function $P_2$ is fitted to the maximal values of the data. The function $P_1$ is then used as a lower boundary for the annual oscillation of a periodic squared sine function, which will take its minimal value provided by the $P_1$ function. The maximal value of the oscillation is given by the $P_2$ function. The functions $P_1$ and $P_2$ determine bounds in which a sine function will be constrained to oscillate to represent the daily average temperature, i. e.

$$T_{out} = P_1 + (P_2 - P_1)Sin\left(\frac{t}{365}\pi\right)^2$$

Where $T_{out}$ will represent the outdoor temperature. In order to manage the complexity of the meteorological time series, we observed that considering the first 12 terms of the polynomial $P = \sum_{n=0}^{12} a_n t^n$, was sufficient for our purposes.

### Precipitation

Similarly, the data are ordered as a time series. For each year, the maximal values are obtained (the minimum being zero). A polynomial function $M_1$ is fitted to the maximal values of the series. As the oscillation of the rainfall R, is not distributed upon time but constrained solely to some months of the year, the oscillation will be modulated by an exponential factor which will produce a thin and stretched oscillation pattern, i. e.

$$R = M_1 Exp\left(10^{-3}Sin\left(\frac{t-30}{365}\pi + 80\right)^2\right)^2$$

Finally, the order of the polynomial functions used will be proportional to the number of years simulated, i. e. the order of the polynomial is equal to the IntegerPart(years simulated /2) + 2.

### Differential equations system

The model considers the aquatic cycle of the mosquito development and its adult cycle, as well as the existence of four different serotypes of Dengue. The model also takes in to account the

stage of latency of eggs during drought season, the Dengue transovarial transmission, time delay for the possibility of transmission of Dengue in both mosquitoes (λ days of delay) and humans (τ days of delay). The variables $G_1$, $G_2$, $G_3$ and $G_4$, represents Dengue-free mosquito's eggs, larva, pupa and adult mosquito, respectively. $G_5$ represents Dengue-free mosquito's eggs latent state during the drought season.

For notation simplification purposes, we use letter S to represent the variables X, Y, Z and W, which represents infection with Dengue serotypes 1, 2, 3 and 4, respectively. The variables $S_1$ represents mosquito eggs infected with Dengue serotype 1, 2, 3 or 4 (depending on if S takes the representation X, Y, Z or W). The variables $S_2$ represents infected larva with the corresponding serotype (again, serotypes 1, 2, 3 and 4 if S takes the representation X, Y, Z and W, in that order), S3 are infected pupa, $S_4$ infected mosquitoes, $S_5$ represents humans infected with serotypes 1, 2, 3 or 4, $S_6$ represents humans immune to infection by DENV serotypes 1, 2, 3 or 4, and finally, $S_7$ represents Dengue's infected eggs in latent state during the drought season. $\dot{X}$ represents the derivative function of time of X, i. e. $\dot{X} = \frac{dX}{dt}$.

$$\dot{G}_1 = k_1\left(G_4(t) + (1 - m)\sum_S S_4(t)\right) + k_{18}G_5(t) - (k_2 + k_{17} + k_3)G_1(t) \tag{1}$$

$$\dot{G}_2 = k_2 G_1(t) - (k_4 + k_5 + k_6 G_2(t))G_2(t) \tag{2}$$

$$\dot{G}_3 = k_5 G_2(t) - (k_7 + k_8)G_3(t) \tag{3}$$

$$\dot{G}_4 = k_7 G_3(t) - \left(k_9 \sum_S S_5(t) + k_{10}\right)G_4(t) \tag{4}$$

$$\dot{G}_5 = k_{17}G_1(t) - (k_{18} + k_{19})G_5(t) \tag{5}$$

$$\dot{S}_1 = mS_5(t) + k_{18}S_7(t) - (k_2 + k_{17} + k_3)S_1(t) \tag{6}$$

$$\dot{S}_2 = k_2 S_1(t) - (k_4 + k_5 + k_6 S_2(t))S_2(t) \tag{7}$$

$$\dot{S}_3 = k_5 S_2(t) - (k_7 + k_8)S_3(t) \tag{8}$$

$$\dot{S}_4 = k_7 S_3(t) + k_9 S_5(t)G_4(t) - k_{11}S_4(t) \tag{9}$$

$$\dot{S}_5 = \frac{k_{12}}{\tau}(H - R(t))S_4(t)\theta(t) - (k_{13} + k_{14})S_5(t) \tag{10}$$

$$\dot{S}_6 = k_{13}S_5(t) - \left(k_{15} + k_{16,S}\right)S_6(t) \tag{11}$$

$$\dot{S}_7 = k_{17}S_1(t) - (k_{18} + k_{19})S_7(t) \tag{12}$$

With $R(t) = S_5(t) + S_6(t) + \Sigma_{N \neq S}(1 - \alpha k_{16,N})N_5(t)$ and $\theta(t) = 1 - \beta\frac{S_5(t) + S_6(t)}{\sum_S S_5(t) + S_6(t)}$, $\Theta$ is the cut-off factor for minimal population proportion that is available to be infected, $\tau$ represents the Extrinsic Incubation Period, which is the time necessary for the mosquito to become infective, $\alpha$ is the relative cross-protection decline rate, and $\beta$ is a calibration parameter, which serves for finding a suitable value (weight) for the fraction of the infected persons of the former serotype,

with respect to all serotypes, in the cut-off function Ө. The value of β was not determined by any optimization algorithm but through a try and failure way.

The included variables (G and S values) and the values and interpretation of the parameters (*k* values) are presented in Table 1. Numerical solutions to the set of equation were obtained using the Mathematica 8 software, with the initial conditions mentioned at Table 1. The program used to solve the equations is provided in the supplemental material (see S1 File).

In the model, indoor and outdoor temperatures are different. This difference is relevant since adult mosquito life is mainly spent indoor, while aquatic stages take place outdoor. The indoor temperatures will be correlated with the outdoors's but with a pull-down factor that will reduce temperature peaks, as observed in the real world [48]. The indoor temperature $T_{in}$ will thus be defined as

$$
T_{in} =
\begin{cases}
T_{min}, & T_{out} < T_{min} \\
T_{out} + \left(1 - 1.3\, e^{-\frac{T_{out} - T_{min}}{2}}\right), & T_{out} \geq T_{min}
\end{cases}
\tag{13}
$$

Where $T_{min}$ is the minimum value allowed for the indoor temperature.

## Scenarios

The validation of the model is achieved through the comparison of the model´s outcomes for the relative prevalence of the Dengue serotypes upon time, with data the data presented by Falcón-Lezama et al., for the consecutives years of 1995–2007 [8]. These data show the relative serotype composition of the epidemic observed in Mexico during the aforementioned period of time. The absolute DENV prevalence values are obtained from the article of Dantés et al. [49], where the epidemiological trends of Dengue in Mexico are estimated. Aside from the DENV cases absolute values of the epidemic, this article also provides the number of hemorrhagic cases presented. In all these works, the original data were collected through the "Centro de Vigilancia Epidemiológica y Control de Enfermedades", the national agency in charge of tracking the diseases spread in Mexico. From the total probable Dengue cases, 30% were sampled for confirmation purposes, as per normative guidelines. The serotype of the circulating virus was determined through PCR genotypification of 10% of the previously confirmed cases [6]. The specific weight of the data obtained by this method was then extrapolated to the observed cases. This method is the one generally used by the Mexican national disease surveillance system to monitor the burden of Dengue in the population.

As a proxy and for simplicity, we will simulate the DENV epidemics in Mexico using the south of Mexico as a national wide reference, due to the fact that the south of Mexico (typically) produces nearly 80% of all the Dengue cases and that all the serotypes are present [50, 51]. For these purposes, we will use the weather pattern of the state of Yucatan as representative of the meteorology of the south of Mexico. Since the peninsular zone of Mexico is a flatland with relatively homogeneous meteorological conditions (including the Pacific coast and the on the shores of the Gulf of Mexico), that that geographical zone is the most likely to present an endemic pattern.

## Results

In the Fig 1, we displayed the temperature (blue: real data, black: outdoor smoothness fit, red: indoor smoothness fit) and the rainfall (blue real data, black outdoor smoothness fit) registered for the Mexican state of Yucatan during the period of 1995–2013. As we previously mentioned, we will use the Yucatan's climatology as reference for our predictions. Considering that the

**Table 1.**

| Variable | Interpretation |
|---|---|
| **Non-infected population** | |
| $G_1$ | Non-infected eggs |
| $G_2$ | Non-infected larvae |
| $G_3$ | Non-infected pupae |
| $G_4$ | Non-infected mosquitoes |
| $G_5$ | Resting non-infected eggs |
| **Infected populations** | |
| $S = X = Serotype\ I;\ S = Y = Serotype\ II;\ S = Z = Serotype\ III;\ S = W = Serotype\ IV$ | |
| $S_1$ | Infected eggs |
| $S_2$ | Infected larvae |
| $S_3$ | Infected pupae |
| $S_4$ | Infected mosquitoes |
| $S_5$ | Infected humans |
| $S_6$ | Immune humans |
| $S_7$ | Resting infected eggs |

**Initial conditions**

$G_1(0) = 0;\ G_2(0) = 0;\ G_3(0) = 0;\ G_4(0) = 1-\mu_m;\ G_5(0) = 0;$

$X_1(0) = 0;\ X_2(0) = 0;\ X_3(0) = 0;\ X_4(0) = f_1\,\mu_m;\ X_5(0) = f_1\,\beta;\ X_6(0) = f_1\,\mu_h;\ X_7(0) = 0$

$Y_1(0) = 0;\ Y_2(0) = 0;\ Y_3(0) = 0;\ Y_4(0) = f_2\,\mu_m;\ Y_5(0) = f_2\,\beta;\ Y_6(0) = f_2\,\mu_h;\ Y_7(0) = 0$

$Z_1(0) = 0;\ Z_2(0) = 0;\ Z_3(0) = 0;\ Z_4(0) = f_3\,\mu_m;\ Z_5(0) = f_3\,\beta;\ Z_6(0) = f_3\,\mu_h;\ Z_7(0) = 0$

$W_1(0) = 0;\ W_2(0) = 0;\ W_3(0) = 0;\ W_4(0) = f_4\,\mu_m;\ W_5(0) = f_4\,\beta;\ W_6(0) = f_4\,\mu_h;\ W_7(0) = 0$

| Parameter | Interpretation | Value | Reference |
|---|---|---|---|
| H | Human density | 3.7 | [28] |
| $k_1(T(t))$ | Mosquito oviposition rate | $k_{12}\,(-71.06 + 7.59\,T_{out} - 0.14\,T_{out}^2)\,/2$ | [29–32] |
| $k_2(T(t))$ | Rate of progression to the larval stage | $(37.06 - 2.08\,T_{out} - 0.03\,T_{out}^2)^{-1}$ | [32] |
| $k_3(T(t))$ | Mortality rate of eggs (during the rainy season) | $0.38\,k_2$ | [32] |
| $k_4(T(t))$ | Rate of progression to the pupal stage | $(55.49 - 2.86\,T_{out} - 0.04\,T_{out}^2)^{-1}$ | [32] |
| $k_5(T(t),R(t))$ | Mortality rate of larvae | $0.25\,\delta\,k_4$ | [32] |
| $k_6$ | Density-dependent mortality rate of larvae | 0.05 | [33] |
| $k_7(T(t),R(t))$ | Rate of progression to mosquito stage | $(18.78 - 1.00\,T_{out} - 0.01\,T_{out}^2)^{-1}$ | [32] |
| $k_8(T(t))$ | Mortality rate of pupae | $0.09\,k_7$ | [32] |
| $k_9$ | Infectious meal rate from humans to mosquitoes | 2.14 | [34, 35] |
| $k_{10}(T(t))$ | Mortality rate of healthy mosquitoes | $(-90.76 - 9.54\,T_{out} - 0.18\,T_{out}^2)^{-1}$ | [32] |
| $k_{11}(T(t))$ | Mortality rate of infected mosquitoes | $1.56\,k_{10}$ | [1] |
| $k_{12}(T(t))$ | Infectious bite rate from mosquitoes to humans | $0.124\,k_{11}$ | [30, 32, 36] |
| $k_{13}$ | Infected human death rate | $0.99\,k_{15}$ | [37, 38] |
| $k_{14}$ | Immunity acquisition rate | 0.14 | [35] |
| $k_{15}$ | Human death rate | $6.5\ 10^{-7}$ | [30] |
| $k_{16}$ | Immunity loss rate | $4.5\ 10^{-4}$ | [39] |
| $k_{16,Y}$ | Immunity loss rate (serotype 2) | $4.5\ 10^{-4}$ | [40] |
| $k_{16,X}$ | Immunity loss rate (serotype 1) | $0.9\,k_{16,Y}$ | [8] |
| $k_{16,Z}$ | Immunity loss rate (serotype 3) | $0.45\,k_{16,Y}$ | [8] |
| $k_{16,W}$ | Immunity loss rate (serotype 4) | $0.07\,k_{16,Y}$ | [8] |
| $k_{17}(R(t))$ | Drought induced egg eclosion inhibition Mosquito emergence deactivation by drought | $1 - k_{18}$ | Supposed |
| $k_{18}(R(t))$ | Rain induced egg eclosion rate | HeavisideTheta $(R - 1)$ | Supposed |

(*Continued*)

**Table 1.** (Continued)

| | | | |
|---|---|---|---|
| $k_{19}$ | Mortality rate of eggs during drought | 0.018 | [41] |
| $\tau(T(t))$ | Extrinsic Incubation Period in the mosquito | $600\,(0.3/2\pi)^{1/2}\,\mathrm{Exp}(-0.3\,(T_{in}-5.9)^2/T_{in})$ | [42] |
| $\lambda$ | Dengue incubation period in humans | 3 | [35] |
| $\delta(R(t))$ | Rainfall-dependent ponderations | $1-(0.1389-0.0136\,R)$ | [43] |
| m | Transovarial transmission | 0.01 | [44] |
| $\mu_m$ | Initial prevalence of DENV in mosquitoes | 0.03 | Publication in press |
| $\mu_h$ | Initial seroprevalence of DENV in humans (immunity) | 0.335 | [45] |
| $\alpha$ | Relative cross-protection decline rate (related to the serotype-specific immunity) | 2 | [40, 46] |
| $\beta$ | Initial incidence of DENV in humans | 0.12 | Calibration parameter |
| $f_1, f_2, f_3, f_4,$ | Proportion of initial serotypes for DENV 1, 2 3 and 4, respectively | 0.6; 0.2; 0.003; 0.197 | [8] |
| | Probability of symptomatic Dengue for serotypes I, II, III and IV, respectably | 0.419; 0.126; 0.169; 0.196 | [47] |
| | Probability of Dengue hemorrhagic fever for serotypes I, II, III and IV, respectably | 0.18; 0.19; 0.13; 0.18 | [12] |

rainfall data are noisier, its smoothing was successful. Only the highest rainfall values failed to be adjusted accurately. Nevertheless, this imprecision will not produce a major deviation in the predictions, since there is a cut-off for the transmission for rainfall >20 mm, as mentioned in reference [43]. The smoothness adjustment of the temperature data does behave correctly in most of the time frame explored, only failing to follow the real temperature data in the years staging the minimum values (years 2004, 2005 and 2012).

The incidence of each Dengue serotype is presented in Fig 2A. As can be observed, the dynamics of DENV serotype 4 (green line) prevalence shows a rapid decrease in the first years of the model prediction. Between the years 1995 and 1999, DENV serotype 3 is the most prevalent (blue line). DENV serotype 2 (red line) presents a notable emergence in 1999 and increases its prevalence (and actually dominates) from the year 2000 to 2005. Also, DENV serotype 2 shows a notoriously minor infection rate than serotype 3. The intrinsic transmission capabilities of serotype 2, the relative epidemiological situation of the other serotypes and importantly the diminution of rainfall observed during the period explain these results.

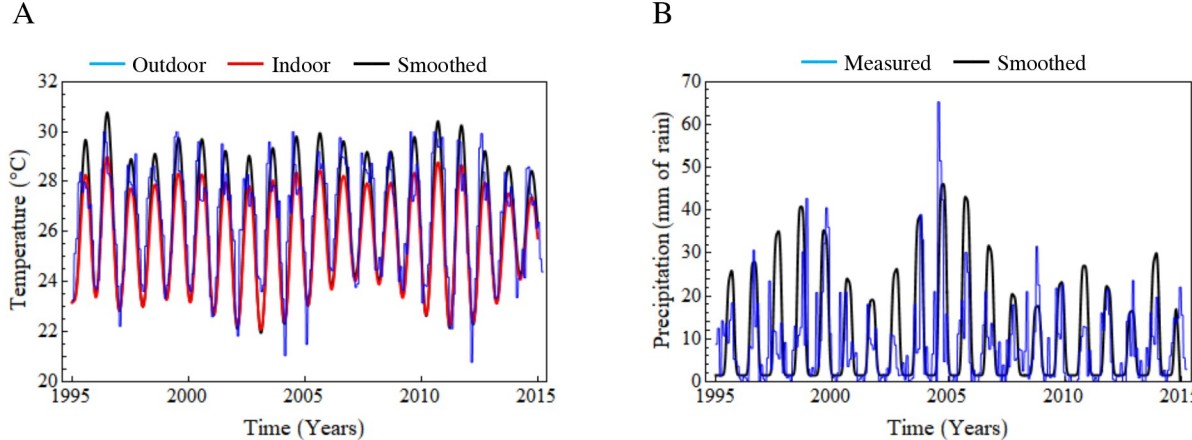

**Fig 1. Reported and adjusted temperature and precipitation data used in the mathematical modeling of the DENV serotypes transmission.** 1A blue line: daily temperature registered, red line: smoothed fit adjusted outdoor temperature, black line smoothed fit adjusted indoor temperature. 1B blue line: daily precipitation registered (mm of rain). Black line smoothed fit adjusted precipitation.

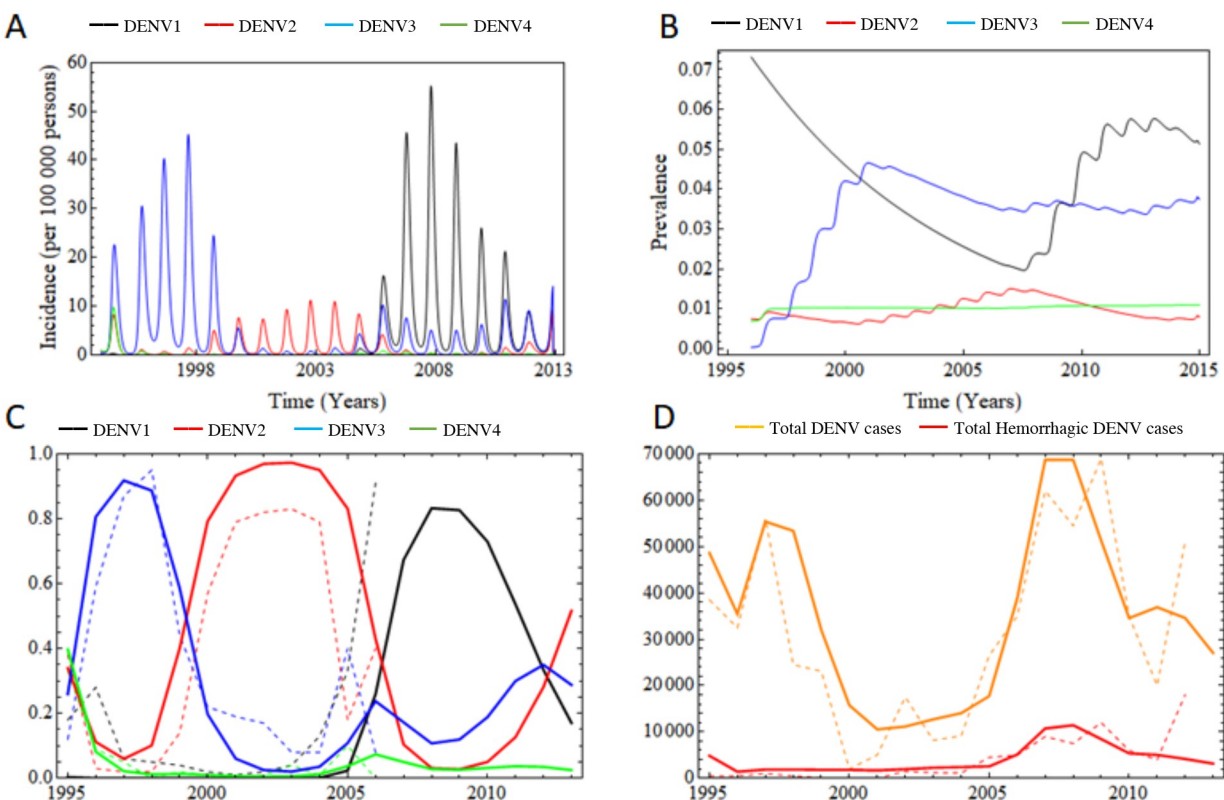

**Fig 2.** 2A predicted incidence of each Dengue serotype: black (DENV1), red (DENV2), blue (DENV3) and green (DENV4). 2B predicted prevalence, calculated as the cumulative incidence of each Dengue serotype: black (DENV1), red (DENV2), blue (DENV3) and green (DENV4). 2C Serotype distribution over time: dotted line (observed data), continuous line (predicted distribution), black (DENV1), red (DENV2), blue (DENV3) and green (DENV4). 2D Yearly cumulative symptomatic cases: orange (total cases), red (total hemorrhagic cases), doted lines (observed cases), continuous line (predicted cases).

Between 2006 to 2011, the DENV serotype 1 (black line) leads the infection dynamic and becomes the dominant serotype. It produces a greater infection peak than the one produced in 1998 by serotype 3. During the 2006–2011 period, serotype 3 prevalence is constant and relevant.

In Fig 2B, we show the seroprevalence of DENV serotype immunity in the population. In this figure, we observe that the cumulated incidence and the immunological protection loss rate parameters, derive in dominant serotype 3 specific immunological protection. The serotype 3 immunity prevalence remains relatively constant from the year 2000. Serotype 1 immunity turn dominant during the year 2009 but show no steady behavior. We can also observe that serotypes 2 and 4 are relatively constant during time in the period 1995–2013. There were no experimental data available for comparison.

The serotype composition of the prevalence was obtained from the normalization of the outcomes of the sum of the variables $S_5(t) + S_6(t)$, which are the variables that contains the information of the total (incident + prevalent) cases, is presented in the Fig 2C. At the first years of the simulation, the serotype 3 DENV is the most prevalent serotype, followed by the DENV serotype 2 which is the following dominant circulating serotype for 6 years, approximately. From the year 2006 on, the most represented serotype is serotype 1, replacing serotype 2. The other serotypes present a complex mixture of increases and decreases of relative prevalence. Comparing the data given by the mathematical model (solid line) with the observed

DENV prevalence data from the period 1995 to 2007 (dotted lines), we observe a good correspondence between them.

Fig 2D represents the cumulative symptomatic observed cases for all serotypes. The orange lines represent the total symptomatic cases, whereas the red lines represent the total hemorrhagic cases. Since the peninsular region produces the major proportion of cases, we are modeling its epidemiological situation (rainfall, temperatures and initial settings) as a proxy of the nationwide epidemiology. The number of cases calculated based on the pluviometry and temperature data of the peninsula region are compared with the data observed nationwide. Since this region amount for most of the cases, we assume that the epidemiological dynamic that we simulate with these data is similar to the one observed nationwide. The model correctly predicts the epidemiological peaks and the epidemiological minimum; though not exactly. The prediction is sufficiently accurate as to be taken into account for local short-term simulations (years) of dengue spread and prevalence.

The accuracy of mathematical modeling to predict the prevalence and serotype of dengue infection over 18 years is tied to the availability of precise weather data and initial conditions of the region to be studied. That being considered, the data obtained from the calculation of the presented model were validated by epidemiological data of the Mexico in a retrospective way, presenting a root-mean-square deviation of 33.9% per year (and 19.8%, dropping the outlier years 1998 and 2008), in the prediction accuracy of the total infection cases.

## Discussion

We found that the model adjusts satisfactorily to the observed data of dispersion and intensity of DENV infection in function of the climate data of specific regions constitutes a definitive instrument to design and perform public health intervention in Mexico. Our former model achieved forecast of a DENV dissemination using a generic DENV virus set of infective characteristics. Since the four serotypes of DENV present differences in infection dynamics, we modeled the spread of each virus and their interaction over 18-year span time.

The model shows that, even though some specific set of initial immunological and climatological conditions would favor one serotype over another, the varying weather and immunological state of the human population ultimately create a succession of different serotypes emergence over time. Furthermore, the differences in immunity generated by the infection from one or another genotype could be related with potentially life-threatening clinical outcome [52]. The prediction capacity of our model gives a root-mean-square deviation of 33.9% per year (19.8%, dropping the outlier years), in the prediction accuracy of the total infection cases. There are some example of statistical models which depends of meteorological variables that produces more accurate predictions, like the work published by Ling Hii et al. [53], which reports an accuracy of 0.3% in a 16-weeks forecasting, or the model published by Naher et al. [54], which reports an accuracy fit of 10.8% in a 2-year time-series model. Nevertheless, our model predicts proportion of DENV serotype as well as the total amount of DENV infection over a 18-year time-lapse, with an accuracy allowing for long-term public health decision.

The model hereby presented cannot predict the interventions that would modify the biological condition in which the mosquito and the human lives. Insecticide and larvicide spraying/application cannot therefor be considered. The impact of these events on the mosquito life/biology tend to be very local and their effect on the natural DENV transmission short lasted [55, 56]. On the other side, drastic climate change would affect transmission through higher mosquito mortality and changes in the time required for the mosquito to be infective.

Finally, we would like to emphasize that the accuracy of mathematical modeling to predict the prevalence and serotype of dengue infection over 18 years is tied to the availability of

precise weather data and initial conditions of the region to be studied, thus the production of a realistic and well delimited scenario is problematic due to the lack of time, geographical and meteorological dependent parameters, but the presented model can be optimized for specific applications.

## Conclusions

The accuracy of the DENV serotype succession and infection intensity predictions of the hereby presented model was tested against the epidemiological data of Lezama et al. reported between years of 1995–2007. In our model, we determined that the succession of the different serotype is driven by the duration of the human serotype specific immunity. The differences of cross specificity between the DENV serotypes induced immunities as well as the specific virulence factors generates the virus alternance observed in the model. In particular, the DENV1 resurgence can be linked to more symptomatic patients. Such prediction would compel to increased containment measures on the field in order to tame the aftermath of the infection wave.

## Supporting information

**S1 File. Code of the program.**
(DOCX)

## Author Contributions

**Conceptualization:** Gilberto Sánchez-González, Renaud Condé.

**Data curation:** Gilberto Sánchez-González.

**Formal analysis:** Gilberto Sánchez-González.

**Methodology:** Gilberto Sánchez-González, Renaud Condé.

**Writing – original draft:** Renaud Condé.

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
