## [Decision Letter · Decision Letter 0]

13 Dec 2022

PONE-D-22-28107Mathematical modeling of Dengue virus serotypes dispersalPLOS ONE

Dear Dr. Condé,

Thank you for submitting your manuscript to PLOS ONE. After careful consideration, we feel that it has merit but does not fully meet PLOS ONE’s publication criteria as it currently stands. Therefore, we invite you to submit a revised version of the manuscript that addresses the points raised during the review process.

We look forward to receiving your revised manuscript.

Kind regards,

Jan Rychtář

Academic Editor

PLOS ONE

Journal Requirements:

    "NO authors have competing interests"

5. Please amend the manuscript submission data (via Edit Submission) to include author Gilberto Sánchez González.

6. Please amend your list of authors on the manuscript to ensure that each author is linked to an affiliation. Authors’ affiliations should reflect the institution where the work was done (if authors moved subsequently, you can also list the new affiliation stating “current affiliation:….” as necessary).

Additional Editor Comments:

The article was reviewed by two expert reviewers.

Reviewer #2 in particular raises several major issues and in summary thinks that it very difficult to understand the model, and believes that the interpretation of the results needs quite a bit of work as well.

The paper is thus on the boundary between major revision and reject. While I am recommending major revision, the authors should be warned that if the issues raised by the reviewers are not addressed properly, the paper can and will be rejected.

Reviewers' comments:

Reviewer's Responses to Questions

**Comments to the Author**

1. Is the manuscript technically sound, and do the data support the conclusions?

Reviewer #1: Yes

Reviewer #2: Partly

2. Has the statistical analysis been performed appropriately and rigorously? 

Reviewer #1: I Don't Know

Reviewer #2: N/A

3. Have the authors made all data underlying the findings in their manuscript fully available?

Reviewer #1: Yes

Reviewer #2: Yes

4. Is the manuscript presented in an intelligible fashion and written in standard English?

Reviewer #1: Yes

Reviewer #2: Yes

5. Review Comments to the Author

Reviewer #1: This is a modelling analysis aiming to determine serotype-specific outbreak probabilities for dengue virus, using 10 years of data and incorporating meteorological information.

I have the following suggestions and requests for clarification:

1. The geographical scope of the analysis is somewhat confusing throughout the text, and greater clarity on whether the data and predictions are national, or targeting the Yuncatán region would be useful.

2. The rationale for smoothing the meteorological data rather than using the raw data were unclear, particularly as from Figure 1A, the raw values in black seem smoother than the smoothed values in blue and red. In addition to this, some more detail in the Discussion section would be useful about the impact of the smoothed values not following the observed temperature data in several years (2004, 2005 and 2012).

3. It is unclear how the model was trained on dengue serotype prevalence data. More detail would be useful on what data is used to inform the model, and whether the model is fit to data or is simulating serotype-specific dynamics based on initial parameters. A section in the methods outlining the data used in the analysis, how this is integrated into the modelling framework and how predictions are generated would be useful.

From the manuscript text and Figure 2C the data time series ends in 2007 with model predictions continuing until 2013. Some detail on why the time series ends there and whether the prediction of a large DENV1 outbreak in subsequent years matched what was observed in Mexico in this period would be useful.

4. Some more information on panels in Figure 2 would also be useful, for instance how the serotype immunity in the population over time is calculated. Axis labels on panels C and D would also be useful to interpret the figure.

5. More detail could be given on how the accuracy of model predictions were assessed.

6. It is unclear to me how serotype-specific climate relationships are included in the modelling framework, for instance were they assumed through fixed model parameters or estimated from the data? It would be useful to see more methodological detail on this and interesting to see results presented on how climatic relationships differ by serotype.

7. In the conclusion the authors conclude that the succession of different serotypes is determined by the duration of human serotype specific immunity. I’m not sure how this follows from the results and discussion earlier in the manuscript where authors focus on differences in the relationship with meteorological variables. More information on this, and how cross-immunity is included in the modelling framework would be useful.

8. In general, more detail in the Discussion section on the limitations of the model and data used would be helpful to interpret the study conclusions.

9. Did the authors consider incorporating uncertainty in model predictions?

10. Is the code for the analysis publicly available?

11. The reference given for the authors’ earlier pan-serotypic model seems to be incorrect.

Reviewer #2: In this manuscript, the authors develop a climate-driven mathematical model of dengue dynamics with the goal of developing a model that can predict dengue outbreaks and, in particular, dengue stereotypes during outbreaks. The model developed is a system of differential equations describing mosquito population and dengue transmission dynamics. The model includes temperature and precipitation in functions for life history characteristics such as mosquito oviposition, development, and survival rates. With this model, the authors make use of meteorological data from the state of Yucatan to predict cases and cases by serotype in the south of Mexico. The authors show that the model replicates, qualitatively, dengue serotype dynamics in Chiapas Mexico over a twenty year period.

The objectives of the work are novel and interesting; however, the authors use a complex model and simplify the presentation of the model in such a way that it is difficult to understand the biological motivation and mathematical implementation of the model to study the questions posed. I think the manuscript would benefit from a significant amount of work to improve the details of the mathematical model and the justifications for the choices made in the model.

Major Comments:

1. A major goal of this work is to predict different serotypes of the virus; however, there seems to be no obvious mechanism driving the different serotypes of the virus. Is this solely based on initial conditions of the serotypes? If so, what are those initial conditions? Cross-immunity is included, but how is longer term immunity included for a given serotype of the virus? How are the probabilities of severe dengue used here given that there are no dynamic processes included for DHF?

2. There were a number of descriptions missing that would help with understanding the model better. For instance, what is m? I could not find a description of this parameter in the manuscript, but it seems to produce new infected mosquito eggs (presumably through vertical transmission) at a rate proportional to the number of Infected Humans in the population. Should this be proportional to the number of infected mosquitoes or is the current form correct? If so, this needs to be explained in more detail.

3. Additionally, the authors mention l and u, but these parameters do not appear in the model?

What does R_s represent biologically? What about \\theta(S5,S6)? And T? I think a more detailed biological explanation of these terms would be helpful.

4. Additionally, if my understanding of the choice of l and u is correct, what is the justification behind choosing different probabilities of biting an infected/immune or susceptible person? Is it expected that infected and immune people have the same level of attraction to mosquitoes but that a susceptible person has a different level?

5. In Line 137 of the model, the authors multiply the incubation period by the rate of infection. First, does tau represent the incubation period or the incubation rate? If it is the latter, I think the form of the model is ok, but the Table should be modified to reflect this. If it is the incubation period, then this choice needs to be explained in more detail, because it is not clear why one would multiply the incubation period with the biting rate.

6. It’s unclear at what rate mosquitoes are born with dengue infection (through vertical transmission). Is this contained within the parameter m in equation S_1 (line 133)? If so, what values were considered for this model? I think m is not in Table 1. In general, the rate of vertical transmission is quite low (see [1]) and many models neglect this altogether in the name of simplicity. I think this model could benefit from excluding vertical transmission if only to simplify the model. Otherwise, could the authors discuss in more detail the reasoning behind the inclusion of vertical transmission of dengue? Additionally, please specify what the vertical transmission rates are and/or what ranges are considered.

7. The qualitative results of the model compared to the data are quite interesting! Overall, the fit to the data seems to be very good. Even quantitatively, the model results do not seem to be very different from the data, which is very impressive.

8. In the introduction, the authors say “We found that the optimal conditions for transmission of each of the four serotypes are very different,” and in the discussion, the authors mention that “…even though some specific set of climatological conditions would favor one serotype over another,….” But this result is not clear from the data or the model output. Is there a way to look at this more closely? For instance, an analysis including a figure such as Figure 2C along with temperature and/or precipitation data that show these relationships more clearly? Would a sensitivity analysis (Even a university analysis) on temperature and/or precipitation show that conditions favor different serotypes? Another way to show this may be consider deviations of temperature and precipitation from average behaviors. Do these deviations impact serotypes different?

9. Another statement in the discussion that “the difference in immunity generated by the infection from one or another serotype can be related to potentially life threading clinical outcome.” Is not entirely clear from the results either. The only significant change in DHF in Figure 2D is around 2007-2009, but this spike is proportional to a significant spike in cases in this time as well. This does seem to correspond to a spike in DENV1 cases during this period. Is it possible to disentangle the relationships between magnitude of cases and immunity from previous outbreaks? Is the thought here that a loss of DENV1 immunity in the population leads to this more severe outbreak of DENV 1 in this time period? Or that two decades of outbreaks of DENV2 and DENV3 have left a population more susceptible to DHF from DENV1? I think it would be helpful to elaborate on this result.

Minor Comments

10. In line 68, the authors mention a previous version of the model, but the reference in (12) is not a modeling work. I’m not sure, but I think the paper the authors meant to cite is [2] below.

11. Line 33: There is an effective dengue vaccine available, although it is not licensed globally. It is, however, licensed in Mexico now. See [3]

12. In line 88, the authors mention P1 and P2 are polynomial functions fitted to values in the data. What do these polynomial functions look like? Can you define these functions in the manuscript as well.

13. In line 137, The model includes \\tau_s but the table includes only \\tau. I think this may be an error, but if not, please explain the difference.

14. Figure 2D: I think these are the “total number of cases of all serotypes” instead of ‘cumulative’ cases. Generally in this context, ‘cumulative’ would imply that cases are added up from the initial time point and would result in a monotonically non-decreasing curve.

Other Comments

15. In the abstract, the first sentence: “The dengue virus (DENV) is the most important mosquito-borne virus worldwide.” ?

16. Line 92: I think rather than ‘define an envelopment’ here, the author’s mean ‘ determine bounds’

17. In Line 102 and in several other places, the author’s have written ‘patron’ where I think they mean ‘pattern’

References

1. Danis‐Lozano, Rogelio, et al. "Vertical transmission of dengue virus in Aedes aegypti and its role in the epidemiological persistence of dengue in Central and Southern Mexico." Tropical Medicine & International Health 24.11 (2019): 1311-1319.

2. Sánchez-González, Gilberto, et al. "Prediction of dengue outbreaks in Mexico based on entomological, meteorological and demographic data." PloS one 13.8 (2018): e0196047.

3. https://www.nature.com/articles/nbt0116-8b

6. PLOS authors have the option to publish the peer review history of their article (what does this mean?). If published, this will include your full peer review and any attached files.

Reviewer #1: No

Reviewer #2: No

---

## [Author Response · Author response to Decision Letter 0]

29 Mar 2023

Reviewer #1: This is a modelling analysis aiming to determine serotype-specific outbreak probabilities for dengue virus, using 10 years of data and incorporating meteorological information.

I have the following suggestions and requests for clarification:

1. The geographical scope of the analysis is somewhat confusing throughout the text, and greater clarity on whether the data and predictions are national or targeting the Yucatan region would be useful.

Resp: Thank you very much for the observation. In our model we used national DENV data, but it is important to underline the fact that 

 during the time period used for the DENV serotypes dispersion simulation, most of the national infections proceeded from the south of Mexico.

 the peninsular weather is very homogeneous in the south of Mexico.

Thus, we are taking the Yucatan´s weather conditions as a proxy weather for the south of Mexico and the south of Mexico as a proxy of the national epidemics. To underline this, we made the following changes to the text:

Original:

“As a proxy, we simulate the DENV epidemics in the south of Mexico, using the weather patron of the state of Yucatán (a peninsular reference). The south of Mexico contributes to most of the national epidemic cases and typically present all Dengue serotypes. It is important to highlight that the peninsular zone of Mexico is a flatland with relatively homogeneous meteorological conditions.”

Reviewed: From line 361 to 367

“As a proxy and for simplicity, we will simulate the DENV epidemics in Mexico using the south of Mexico as a national wide reference, due to the fact that the south of Mexico (typically) produces nearly 80% of all the Dengue cases and that all the serotypes are present (42), (43). For these purposes, we will use the weather patron of the state of Yucatán as representative of the meteorology of the south of Mexico. Since the peninsular zone of Mexico is a flatland with relatively homogeneous meteorological conditions (including the shores of the Gulf of Mexico), that geographical zone is the most likely to present an endemic pattern.”

42. Dzul-Manzanilla F, Correa-Morales F, Che-Mendoza A, Palacio-Vargas J, Sánchez-Tejeda G, González-Roldan JF, López-Gatell H, Flores-Suárez AE, Gómez-Dantes H, Coelho GE, et al. Identifying urban hotspots of dengue, chikungunya, and Zika transmission in Mexico to support risk stratification efforts: a spatial analysis. Lancet Planet Heal (2021) doi: 10.1016/S2542-5196(21)00030-9

43. Navarrete J, Vázquez ;, Gómez ; Epidemiología del Dengue y Dengue Hemorrágico en el Instituto Mexicano del Seguro Social (IMSS). Rev Peru Epidemiol (2002).”

2. The rationale for smoothing the meteorological data rather than using the raw data were unclear, particularly as from Figure 1A, the raw values in black seem smoother than the smoothed values in blue and red. In addition to this, some more detail in the Discussion section would be useful about the impact of the smoothed values not following the observed temperature data in several years (2004, 2005 and 2012).

Resp: We agree with the revisor that clarifications are needed. Therefore we introduced the next text in the methods section, lines 85 to 87:

“As a first approach, we fed the model with the original discrete values of rainfall and temperature, but we noted the appearance of singularities due to the lack of continuity in those parameters, thus we established the following methodology for the data smoothing:” 

And in the results section, lines 183 to 184:

“Considering that the rainfall data are noisier, its smoothing was successful. Only the highest rainfall values fail to be adjusted accurately Nevertheless, this imprecision will not produce a major deviation in the predictions, since there is a cut-off for the transmission for rainfall >20 mm, as mentioned in reference (39).”

It is unclear how the model was trained on dengue serotype prevalence data. More detail would be useful on what data is used to inform the model, and whether the model is fit to data or is simulating serotype-specific dynamics based on initial parameters. A section in the methods outlining the data used in the analysis, how this is integrated into the modelling framework and how predictions are generated would be useful.

Resp: please note that our approach is deterministic, so no model training is needed. Derived from your question, we noted that we omitted to specify the calculation method. We introduced the next statement:

“Numerical solutions to the set of equation were obtained using the Mathematica 8 software, with the initial conditions mentioned at Table 1. The program used to solve the equations is provided in the supplemental material”

From the manuscript text and Figure 2C the data time series ends in 2007 with model predictions continuing until 2013. Some detail on why the time series ends there and whether the prediction of a large DENV1 outbreak in subsequent years matched what was observed in Mexico in this period would be useful.

Resp: Thank you for the observation. We agree that we have to straighten our graphic presentation. Considering that we do not have statistical data to compare the serotype behavior in the years posterior to 2013, we decided to cut the outcomes of the model to this year. This allow us to produce a major propagation of the calculation error. We will make these changes with no other additional comment in the text.

4. Some more information on panels in Figure 2 would also be useful, for instance how the serotype immunity in the population over time is calculated. Axis labels on panels C and D would also be useful to interpret the figure.

Resp: Thank you very much for your observation. We added more input to the table, including a complete set of initial conditions and their references. We also identified other missing parameters that we integrated in the table. On the other hand, the scale labels of graphs C and D were added to the figure.

5. More detail could be given on how the accuracy of model predictions were assessed.

Resp: As we mentioned at the response of your second question, this is a deterministic model and we do not proceed as is the usage of statistical models where the fit or accuracy of the model is compared to the dependent variable, via a mean-square deviation. In our case, the results shown in the figure 2C and 2D are predictive in the continuous line. The dotted line is the DENV serotypes prevalence observed during the same period, and these data were not taken into account to feed the model. Hence the model is able to predict/reproduce the serotype prevalence using solely the temperature and pluviometry observed during this period, and the initial (1995) DENV conditions. Regarding your commentary, we introduced the following paragraph in the result section:

“The accuracy of mathematical modeling to predict the prevalence and serotype of dengue infection over 18 years is tied to the availability of precise weather data and initial conditions of the region to be studied. That being considered, the data obtained from the calculation of the presented model were validated by epidemiological data of the Mexico in a retrospective way, showing a root-mean-square deviation of 12,690 per year, in the prediction accuracy of the total infection cases.”

And in the discussion section:

“Finally, we would like to emphasize that the accuracy of mathematical modeling to predict the prevalence and serotype of dengue infection over 18 years is tied to the availability of precise weather data and initial conditions of the region to be studied, thus the production of a realistic and well delimited scenario is problematic due to the lack of time, geographical and meteorological dependent parameters, but the presented model can be optimized for specific applications.”

6. It is unclear to me how serotype-specific climate relationships are included in the modelling framework, for instance were they assumed through fixed model parameters or estimated from the data? It would be useful to see more methodological detail on this and interesting to see results presented on how climatic relationships differ by serotype.

We agree with the revisor commentary and apologies for not referring the complete set of parameters used to calculate the outcome of the model. We corrected that omission in the text and in the parameter Table 1. The differences between the serotypes relied fundamentally on the time taken to lose immunity after the infection from one or the other serotype and the initial conditions. Hence no other additional comment was added in the text. 

7. In the conclusion the authors conclude that the succession of different serotypes is determined by the duration of human serotype specific immunity. I’m not sure how this follows from the results and discussion earlier in the manuscript where authors focus on differences in the relationship with meteorological variables. More information on this, and how cross-immunity is included in the modelling framework would be useful.

Resp: Indeed, the results of the modeling showed us that the main factor driving the serotype exchange was the serotype specific immunity in the human population. The differences in the mosquito infection between the different serotypes were not sufficient to influence the serotype dynamics, neither was the different dynamics of virus progression in the human. The model was able to foresee the serotype changes and intensities without taking these parameters into account.

The differences between the serotypes viremia observed in human are of one Log10 , as reported in Comparative susceptibility of aedes albopictus and aedes aegypti to dengue virus infection after feeding on blood of viremic humans: implications o public health , Whitehorn et al 2015, but this difference does not impact significatively the mosquito infection probability. Furthermore, the probability of these mosquito to posteriorly infect a human is unknown. No other additional comment was added in the text.

8. In general, more detail in the Discussion section on the limitations of the model and data used would be helpful to interpret the study conclusions.

Following your suggestions, we added the following paragraph:

Resp: The model hereby presented cannot predict the man-made action that would modify the biological condition in which the mosquito and the human lives. Insecticide and larvicide spraying/application cannot therefor be taken into account. The impact of these measures on the mosquito life/biology tend to be very local and their effect on the natural DENV transmission short lasted (A model of the transmission of dengue fever with an evaluation of the impact of ultra-low volume insecticide application on dengue epidemics, Newton E., 1993)(1the impact of insecticide treated curtains on dengue virus transmission: a cluster randomized trial in Iquitos,Peru Lhenard A. et al. PLoS Negl Trop Dis. 2020 Apr 10;14(4)). On the other side, drastic climate change would affect transmission through higher mosquito mortality and changes in the time required for the mosquito to be infective. (Lines 563 to 568)

9. Did the authors consider incorporating uncertainty in model predictions?

Resp: as we are running a deterministic model, uncertainty component was not considered.

10. Is the code for the analysis publicly available?

Resp: yes, we will produce a supplemental material with the code of the program. This new material was mentioned in the text.

11. The reference given for the authors’ earlier pan-serotypic model seems to be incorrect.

Resp: Thank you very much for the remark.We changed the reference to the correct one: “Sánchez-González G, Condé R, Moreno RN, López Vázquez PC. Prediction of dengue outbreaks in Mexico based on entomological, meteorological and demographic data. PLoS One (2018) doi: 10.1371/journal.pone.0196047”

Reviewer #2: In this manuscript, the authors develop a climate-driven mathematical model of dengue dynamics with the goal of developing a model that can predict dengue outbreaks and, in particular, dengue stereotypes during outbreaks. The model developed is a system of differential equations describing mosquito population and dengue transmission dynamics. The model includes temperature and precipitation in functions for life history characteristics such as mosquito oviposition, development, and survival rates. With this model, the authors make use of meteorological data from the state of Yucatan to predict cases and cases by serotype in the south of Mexico. The authors show that the model replicates, qualitatively, dengue serotype dynamics in Chiapas Mexico over a twenty year period.

The objectives of the work are novel and interesting; however, the authors use a complex model and simplify the presentation of the model in such a way that it is difficult to understand the biological motivation and mathematical implementation of the model to study the questions posed. I think the manuscript would benefit from a significant amount of work to improve the details of the mathematical model and the justifications for the choices made in the model.

Major Comments:

1. A major goal of this work is to predict different serotypes of the virus; however, there seems to be no obvious mechanism driving the different serotypes of the virus. Is this solely based on initial conditions of the serotypes? If so, what are those initial conditions? Cross-immunity is included, but how is longer term immunity included for a given serotype of the virus? How are the probabilities of severe dengue used here given that there are no dynamic processes included for DHF?

Resp: Thank you for your commentary. The main mechanism driving the serotype alternance is indeed the specific and pan serotypic immunity generated in the human population. The initial conditions have been added to the manuscript as part of the Table one in the reviewed manuscript. We also have included the duration of iso serotypic and pan serotypic protection (k16 Y,X,Z and W in table 1) used in the equations of the model. We deeply apologize for this crucial omission, without which the mechanism of the model are difficult to understand.

2. There were a number of descriptions missing that would help with understanding the model better. For instance, what is m? I could not find a description of this parameter in the manuscript, but it seems to produce new infected mosquito eggs (presumably through vertical transmission) at a rate proportional to the number of Infected Humans in the population. Should this be proportional to the number of infected mosquitoes or is the current form correct? If so, this needs to be explained in more detail.

Resp: We thank you for the commentary about the description of M. Indeed, as you mentioned, this factor corresponds to the vertical transmission or transovarial transmission of the virus. This factor was set at 0.01, being a middle ground between the publications cited in the table about this matter. The description of this term is added in the table 1.

3. Additionally, the authors mention l and u, but these parameters do not appear in the model?

What does R_s represent biologically? What about \\theta(S5,S6)? And T? I think a more detailed biological explanation of these terms would be helpful.

R_S=S_5 (t)+S_6 (t)+∑_(N≠S)▒〖(1-2k_(16,N))N_5 (t) 〗

Resp: These terms (I and u) were eliminated from the text and, in effect, were not incorporated in the model. T referred to the sum ∑_S▒〖S_5 (t) 〗+S_6 (t), but it was written explicitly in the equations.

Rs is the number of people that cannot be infected from a determined serotype (the subscript s). This correspond to the sum of infected and immune humans from a specific serotype at any time (t) plus the crossed immunity given by the immunity to the other serotypes; hence the sum of the crossed immunity protection for the 3 other serotypes immunity.

Tetha (S5,S6) is the cut off factor for minimal population proportion that is available to be infected. The description of this factor is added at line 145 of the manuscript. We made a simplification of the notation to this version of the manuscript.

4. Additionally, if my understanding of the choice of l and u is correct, what is the justification behind choosing different probabilities of biting an infected/immune or susceptible person? Is it expected that infected and immune people have the same level of attraction to mosquitoes but that a susceptible person has a different level?

Resp: Initially l and u were introduced as factor modulating the biting proportions, but in the preliminary result of the model we observed that this factor was irrelevant for the computed results. As such we took them out. We apologies for this overlook and the inconvenience resulting from this.

5. In Line 137 of the model, the authors multiply the incubation period by the rate of infection. First, does tau represent the incubation period or the incubation rate? If it is the latter, I think the form of the model is ok, but the Table should be modified to reflect this. If it is the incubation period, then this choice needs to be explained in more detail, because it is not clear why one would multiply the incubation period with the biting rate.

Resp: Tau represent the EIP: Extrinsic Incubation Period, which is the time necessary for the mosquito to become infective. The duration of the EIP is essentially dependent on the temperature at which the mosquito is reared. We used the temperature of the inside of the house because that would be where the fed female mosquito would stay most of the time. We changed EIP achronym to Extrinsic Incubation Period in the mosquito in the variable table. Indeed, the Tau parameter had to appear with a (-1) exponent, and instead appeared with an S subindex. We realized that some of the manuscript´s formats was modified when it was open in a Mac computer. We managed to fix the problem and changed the notation to avoid the Macintosh format error. Many thanks. 

6. It’s unclear at what rate mosquitoes are born with dengue infection (through vertical transmission). Is this contained within the parameter m in equation S_1 (line 133)? If so, what values were considered for this model? I think m is not in Table 1. In general, the rate of vertical transmission is quite low (see [1]) and many models neglect this altogether in the name of simplicity. I think this model could benefit from excluding vertical transmission if only to simplify the model. Otherwise, could the authors discuss in more detail the reasoning behind the inclusion of vertical transmission of dengue? Additionally, please specify what the vertical transmission rates are and/or what ranges are considered.

Resp: Indeed, we omitted a correct explanation for the vertical transmission factor m. We added the proper explanation of this factor in table 1. The transmission factor chosen was of 0.01. As reviser 2 aptly notice, the impact of the vertical transmission is very low and is widely outset by the rate of natural transmission. In order to have an immediate effect on the DENV dynamics it should be above 10 %. Nevertheless, considering this DENV transmission via allows for resurgence of the disease in area were the virus is not transmitted year long, and that have lapses of DENV free season.

7. The qualitative results of the model compared to the data are quite interesting! Overall, the fit to the data seems to be very good. Even quantitatively, the model results do not seem to be very different from the data, which is very impressive.

Resp: Thank you for your animating comment. 

8. In the introduction, the authors say “We found that the optimal conditions for transmission of each of the four serotypes are very different,” and in the discussion, the authors mention that “…even though some specific set of climatological conditions would favor one serotype over another,….” But this result is not clear from the data or the model output. Is there a way to look at this more closely? For instance, an analysis including a figure such as Figure 2C along with temperature and/or precipitation data that show these relationships more clearly? Would a sensitivity analysis (Even a university analysis) on temperature and/or precipitation show that conditions favor different serotypes? Another way to show this may be consider deviations of temperature and precipitation from average behaviors. Do these deviations impact serotypes different?

Resp: Indeed your statement is accurate. We wrote it referring to our former model which was temperature and precipitation dependent. In this model we found that the main factor driving the serotype exchange is the human DENV immunity status. Nevertheless, in the literature we found that there are differences between serotype in the mosquito infection rate that are temperature dependent. We agree that our statement in the article is misleading and decided to remove it from the text.

9. Another statement in the discussion that “the difference in immunity generated by the infection from one or another serotype can be related to potentially life threading clinical outcome.” Is not entirely clear from the results either. The only significant change in DHF in Figure 2D is around 2007-2009, but this spike is proportional to a significant spike in cases in this time as well. This does seem to correspond to a spike in DENV1 cases during this period. Is it possible to disentangle the relationships between magnitude of cases and immunity from previous outbreaks? Is the thought here that a loss of DENV1 immunity in the population leads to this more severe outbreak of DENV 1 in this time period? Or that two decades of outbreaks of DENV2 and DENV3 have left a population more susceptible to DHF from DENV1? I think it would be helpful to elaborate on this result.

Resp: Regarding the statement that “the difference in immunity generated by the infection from one or another serotype can be related to potentially life threading clinical outcome.”, we refer it to a finding of other work (we now cites it) and not to a result of our article. In effect, we do not consider mortality in our work. We made the following change to underline this fact: 

“The model shows that, even though some specific set of initial immunological and climatological conditions would favor one serotype over another, the varying weather and immunological state of the human population ultimately create a succession of different serotypes emergence over time. Furthermore, the differences in immunity generated by the infection from one or another genotype could be related with potentially life-threatening clinical outcome (Soo KM, Khalid B, Ching SM, Chee HY. Meta-analysis of dengue severity during infection by different dengue virus serotypes in primary and secondary infections. PLoS One (2016) doi: 10.1371/journal.pone.0154760).”

For the question that if we can disentangle the relationships between magnitude of cases and immunity from previous outbreaks, we want to emphasize that the proportion of iso immunity and pan immunity applied to the model cannot be separated. Moreover, since the observed cases are distinct from the total cases (symptomatic plus asymptomatic) for each serotype, we cannot determine the cumulated iso immunity from the pan immunity protection. 

Minor Comments

10. In line 68, the authors mention a previous version of the model, but the reference in (12) is not a modeling work. I’m not sure, but I think the paper the authors meant to cite is [2] below.

Resp: Done, indeed we misplaced this citation; we thank you for the observation and changed to the correct one (Sánchez-González G, Condé R, Moreno RN, López Vázquez PC. Prediction of dengue outbreaks in Mexico based on entomological, meteorological and demographic data. PLoS One (2018) doi: 10.1371/journal.pone.0196047).

11. Line 33: There is an effective dengue vaccine available, although it is not licensed globally. It is, however, licensed in Mexico now. See [3]

Resp: Indeed the vaccine dengvaxia from Sanofi-pasteur has been licensed for its use in Mexico, though some authors expressed doubt about the vaccine’s prospects : “a 20,000-patient trial in Central and South America resulted in 50%, 74% and 77% protection against serotypes 1, 3 and 4, respectively, but only 35% against serotype “(Nat Biotechnol. 2016 Jan;34(1):8. doi: 10.1038/nbt0116-8b.Mexico dengue vaccine first). Furthermore, though the protection is good the vaccine has shown an increased risk of severe dengue in seronegative subjects [Villar, L.; Dayan, G.H.; Arredondo-García, J.L.; Rivera, D.M.; Cunha, R.; Deseda, C.; Reynales, H.; Costa, M.S.; Morales-Ramírez, J.O.; Carrasquilla, G.; et al. Efficacy of a Tetravalent Dengue Vaccine in Children in Latin America. N. Engl. J. Med. 2015, 372, 113–123]. We corrected our assertion about DENV vaccine availability from line 42 to 46 as following:

“The Dengue vaccine available so far in Mexico display immunity between 35 and 72 %, depending on the serotype (Nat Biotechnol. 2016 Jan;34(1):8. doi: 10.1038/nbt0116-8b. Mexico dengue vaccine first) and present an increased risk of severe dengue in seronegative subjects (Villar, L.; Dayan, G.H.; Arredondo-García, J.L.; Rivera, D.M.; Cunha, R.; Deseda, C.; Reynales, H.; Costa, M.S.; Morales-Ramírez, J.O.; Carrasquilla, G.; et al. Efficacy of a Tetravalent Dengue Vaccine in Children in Latin America. N. Engl. J. Med. 2015, 372, 113–123). Therefor the most commonly available mitigation strategies are the vector control and the prevention of extensive exposure of the population to potential Dengue vectors (through the use of bed nets, fumigation and repellents) (hee-Fu Yung, Kim-Sung Lee, Tun-Linn Thein, Li-Kiang Tan, Victor C. Gan, Joshua G. X. Wong, David C. Lye, Lee-Ching Ng and Y-SL. Dengue Serotype-Specific Differences in Clinical Manifestation, Laboratory Parameters and Risk of Severe Disease in Adults, Singapore. Am J Trop Med Hyg (2015) 92:999–1005. doi: 10.4269/ajtmh.14-0628).”

12. In line 88, the authors mention P1 and P2 are polynomial functions fitted to values in the data. What do these polynomial functions look like? Can you define these functions in the manuscript as well.

Resp: We thank reviser two for pointing out this omission. We added the following statement to the manuscript: “In order to manage the complexity of the meteorological time series, we observed that considering the first 12 terms of the polynomial P=∑_(n=0)^12▒〖a_n t^n 〗, was sufficient for our purposes.”, thereby defining the missing function.

13. In line 137, The model includes \\tau_s but the table includes only \\tau. I think this may be an error, but if not, please explain the difference.

Resp: Thank you for the observation, we corrected the notation, and left tau without subindex.

14. Figure 2D: I think these are the “total number of cases of all serotypes” instead of ‘cumulative’ cases. Generally in this context, ‘cumulative’ would imply that cases are added up from the initial time point and would result in a monotonically non-decreasing curve.

Resp: We changed the graphic to Cases and add a description of the Y axis of figure 2D in the text as Yearly accumulated symptomatic cases.

Other Comments

15. In the abstract, the first sentence: “The dengue virus (DENV) is the most important mosquito-borne virus worldwide.” ?

Resp: we changed the wording to: “The Dengue virus (DENV) constitutes a major vector borne virus disease worldwide.”

16. Line 92: I think rather than ‘define an envelopment’ here, the author’s mean ‘ determine bounds’

Resp: We changed the sentence as suggested. 

17. In Line 102 and in several other places, the author’s have written ‘patron’ where I think they mean ‘pattern’

Resp: We changed the term patron to pattern as suggested on the lines 109 and 177. We thank U form the commentary.

References

1. Danis‐Lozano, Rogelio, et al. "Vertical transmission of dengue virus in Aedes aegypti and its role in the epidemiological persistence of dengue in Central and Southern Mexico." Tropical Medicine & International Health 24.11 (2019): 1311-1319.

2. Sánchez-González, Gilberto, et al. "Prediction of dengue outbreaks in Mexico based on entomological, meteorological and demographic data." PloS one 13.8 (2018): e0196047.

3. https://www.nature.com/articles/nbt0116-8b

---

## [Decision Letter · Decision Letter 1]

2 May 2023

PONE-D-22-28107R1Mathematical modeling of Dengue virus serotypes dispersalPLOS ONE

Dear Dr. Condé,

Thank you for submitting your manuscript to PLOS ONE. After careful consideration, we feel that it has merit but does not fully meet PLOS ONE’s publication criteria as it currently stands. Therefore, we invite you to submit a revised version of the manuscript that addresses the points raised during the review process.

The reviewer was happy with the revision and raises only a couple small points. The manuscript should be acceptable once those are addressed.==============================

We look forward to receiving your revised manuscript.

Kind regards,

Jan Rychtář

Academic Editor

PLOS ONE

Journal Requirements:

Reviewers' comments:

Reviewer's Responses to Questions

**Comments to the Author**

1. If the authors have adequately addressed your comments raised in a previous round of review and you feel that this manuscript is now acceptable for publication, you may indicate that here to bypass the “Comments to the Author” section, enter your conflict of interest statement in the “Confidential to Editor” section, and submit your "Accept" recommendation.

Reviewer #1: (No Response)

2. Is the manuscript technically sound, and do the data support the conclusions?

Reviewer #1: Yes

3. Has the statistical analysis been performed appropriately and rigorously? 

Reviewer #1: Yes

4. Have the authors made all data underlying the findings in their manuscript fully available?

Reviewer #1: Yes

5. Is the manuscript presented in an intelligible fashion and written in standard English?

Reviewer #1: Yes

6. Review Comments to the Author

Reviewer #1: The revised version of this manuscript is much clearer, (particularly the details of the mathematical model used), and as a result the interesting findings on serotype-specific dengue prediction are much more compelling.

Considering the responses to reviewer comments provided, and the new methodological details provided I have the following suggestions:

1. Some specific background to the dengue burden and epidemiology in Mexico would be useful in the introduction, particularly with reference to some of the findings from earlier publications later used as model parameters (e.g. DENV1-4 prevalence over time).

2. In line 14 the authors state that they “used a mathematical model to determine the outbreak probability of specific serotypes of DENV” as no outbreak probabilities are calculated in this paper I would suggest making the aims clearer and more related to the final analysis (for instance, ‘predicting serotype-specific DENV prevalence and overall case burden in Mexico’).

3. Methods: the model equations in the revised text are much clearer.

- Parameter beta is described as a calibration parameter - some description of how calibration was performed would be useful in the main text.

- The revised equations on lines 133 - 147 are substantially clearer. To improve the readability further it would be useful if a consistent subscriptwas used for parameters which are serotype-dependent.

- It would be useful to define theta in the the parameter table as well as the main text.

- In the parameter table I believe P(t) is used to refer to precipitation - I would recommend altering this to read Precipitation(t) or similar to avoid confusion with probability.

- The terminology ‘mosquito emergence deactivation’ and ‘mosquito emergence activation’ is not very clear, I would recommend renaming these perhaps to ‘rate of mosquito emergence’ and ‘rate of mosquito emergence suppression by drought’ or similar.

- I think there is an error in the parameter table for the proportion of initial serotypes for DENV 1,2,3 and 4 as the value for DEV 4 reads 19.997. Should this be 0.19997? If not I’m unsure why these would not sum to 1. It would also be useful to give some context on the data / literature informing this parameter in the introduction as it is quite important context to interpret the subsequent serotype-specific results.

- In line 162 authors describe how the indoor temperature is modelled with a pull-down factor reducing the peaks, this makes sense but differs from what is shown in Fig 1.A where the indoor temperature (black) shows higher peaks than the outdoor temperature (red)?

- It would be useful to add legends to all figures, particularly the serotype specific plots, for ease of interpretation.

4. It’s not clear to me how the initial proportion of serotypes shown in Fig 2.C corresponds with the proportion of initial serotypes for DEV1,2,3 and 4 in the parameter table.

5. More detail on the source and limitations of the data streams compared with model output would be useful - some details on the DENV prevalence data from 1995-2007 and the surveillance used would help readers to qualitatively interpret model fit to data. This could perhaps be included in a data sub-section within the Methods.

6. The model is able to qualitatively capture both the serotype-specific and overall case dynamics (shown in Fig 2.C) which is very interesting. The RMSE is also calculated to assess model accuracy. I would be interested in seeing how this compares to some kind of baseline model (perhaps a simple seasonality / climatological model) although this would depend on the time units of the model presented here.

7. It’s still unclear to me the way in which the different serotypes are differentially affected by temperature as mentioned in line 267, some more discussion on this would be helpful.

Minor comments:

- The use of the term ‘dispersal’ in the title is confusing as it suggests a spatial analysis, I would recommend different terminology used by authors later on in the text such as dynamics or prevalence.

- In line 5 I think rather than ‘constitute’ the authors mean ‘constitutes’.

- In line 33 authors refer to the dengue vaccine displaying ‘immunity’ but I believe ‘protection’ or ‘efficacy’ would be less confusing.

- In line 164 “were” should be replaced with “where”

- In line 184 the sentence “we graphed the temperature” could be replaced with “we displayed the temperature”

- In line 274 the term “man made action” is unclear and could be replaced by “public health interventions”.

7. PLOS authors have the option to publish the peer review history of their article (what does this mean?). If published, this will include your full peer review and any attached files.

Reviewer #1: No

---

## [Author Response · Author response to Decision Letter 1]

23 Jun 2023

1. Some specific background to the dengue burden and epidemiology in Mexico would be useful in the introduction, particularly with reference to some of the findings from earlier publications later used as model parameters (e.g. DENV1-4 prevalence over time).

Resp: We introduced the next paragraph in the introduction section:

“The south-southeast regions of Mexico are endemic for dengue, showing incidence variations depending on the specific weather condition. The magnitude of the epidemic outbreaks also depends on the circulating serotypes. As a consequence, a dengue epidemiological surveillance of the 32 states of the Mexican Republic is performed by the General Directorate of Epidemiology (DGE) [6]. Even though prevention and vector control efforts have been ramped up, the Dengue cases have been multiplied by ten during the last 20 years, reaching 264,898 cases in 2019 [6]. It is worth mentioning that it has been stated that underreporting and misdiagnosis of the DENV cases limit the ability to determine the true burden of disease [7].”

2. In line 14 the authors state that they “used a mathematical model to determine the outbreak probability of specific serotypes of DENV” as no outbreak probabilities are calculated in this paper I would suggest making the aims clearer and more related to the final analysis (for instance, ‘predicting serotype-specific DENV prevalence and overall case burden in Mexico’).

Resp: thank you very much for the suggestion, we changed the sentence to: “Here, we used a mathematical model to predict serotype-specific DENV prevalence and overall case burden in Mexico.”

3. Methods: the model equations in the revised text are much clearer.

- Parameter beta is described as a calibration parameter - some description of how calibration was performed would be useful in the main text.

Resp: we included the following mention in the methods section: “β is a calibration parameter, which serves for finding a suitable value (weight) for the fraction of the infected persons of the former serotype, with respect to all serotypes, in the cut-off function ϴ. The value of β was not determined by any optimization algorithm but through a try and failure way.”

- The revised equations on lines 133 - 147 are substantially clearer. To improve the readability further it would be useful if a consistent subscript was used for parameters which are serotype-dependent.

Resp: only the parameters k_16 are serotype-dependent, and they carry a subscript which define the serotypes referred to.

- It would be useful to define theta in the parameter table as well as the main text.

Resp: Good observation, the theta of the table is the Heaviside Theta, which is now clarified in the text.

- In the parameter table I believe P(t) is used to refer to precipitation - I would recommend altering this to read Precipitation(t) or similar to avoid confusion with probability.

Resp: we changed the name of the variable for R(t) to keep in mind that it refers to rainfall.

- The terminology ‘mosquito emergence deactivation’ and ‘mosquito emergence activation’ is not very clear, I would recommend renaming these perhaps to ‘rate of mosquito emergence’ and ‘rate of mosquito emergence suppression by drought’ or similar.

Resp: we will use the term: Drought induced egg eclosion inhibition and Rain induced egg eclosion rate.

- I think there is an error in the parameter table for the proportion of initial serotypes for DENV 1,2,3 and 4 as the value for DEV 4 reads 19.997. Should this be 0.19997? If not I’m unsure why these would not sum to 1. It would also be useful to give some context on the data / literature informing this parameter in the introduction as it is quite important context to interpret the subsequent serotype-specific results.

Resp: Thank you very much, it was a typographic error, the correct number is 0.197.

- In line 162 authors describe how the indoor temperature is modelled with a pull-down factor reducing the peaks, this makes sense but differs from what is shown in Fig 1.A where the indoor temperature (black) shows higher peaks than the outdoor temperature (red)?

Resp: The black line represents the adjusted curve. In the actualized version of the figures we clarify this point.

- It would be useful to add legends to all figures, particularly the serotype specific plots, for ease of interpretation.

Resp: Done

4. It’s not clear to me how the initial proportion of serotypes shown in Fig 2.C corresponds with the proportion of initial serotypes for DEV1,2,3 and 4 in the parameter table.

Resp: Please consider that the figure is obtained from the normalization of the outcomes, showing the relative fraction of each serotype. More precisely, the graph is showing the normalization on the sum of the variables S_5 +S_6, hence the initial conditions cannot be seen in a strict way. This is now addressed in the text. We introduced the following phrase: “The serotype composition of the prevalence was obtained from the normalization of the outcomes of the sum of the variables S_5 (t)+S_6 (t), which are the variables that contains the information of the total (incident + prevalent) cases, is presented in the Figure 2C”

5. More detail on the source and limitations of the data streams compared with model output would be useful - some details on the DENV prevalence data from 1995-2007 and the surveillance used would help readers to qualitatively interpret model fit to data. This could perhaps be included in a data sub-section within the Methods.

Resp: The prevalence and serotypes data were originally collected through the “Centro Nacional de Vigilancia Epidemiológica y Control de Enfermedades”, the mexican national agency in charge of tracking the diseases spread in Mexico, as well as determine the opportune action to be had. The original data are the results of sampling of the total DENV cases where the serotype was determined by PCR. Since only a fraction of the total DENV cases detected were serotyped, the authors extrapolate the serotype proportion to the total DENV cases encountered in the endemic region. The authors mention the caveat of this generalization and the possibility of having a heterogeneous sampling in the population. Nevertheless, the sum of the 4 serotypes do follow the trend of the total dengue cases reported, hence we assumed that the reported proportion of each serotypes is accurate.

The data can be found at the Dirección General de Epidemiología, Secretaría de Salud. Panorama Epidemiológico (México). México, DF: DGE/SS, 2008. We introduced the next paragraph in the methodology section: “. In all these works, the original data were collected through the “Centro de Vigilancia Epidemiológica y Control de Enfermedades”, the national agency in charge of tracking the diseases spread in Mexico. From the total probable Dengue cases, 30% were sampled for confirmation purposes, as per normative guidelines. The serotype of the circulating virus was determined through PCR genotypification of 10% of the previously confirmed cases [6]. The specific weight of the data obtained by this method was then extrapolated to the observed cases. This method is the one generally used by the Mexican national disease surveillance system to monitor the burden of Dengue in the population.”

6. The model is able to qualitatively capture both the serotype-specific and overall case dynamics (shown in Fig 2.C) which is very interesting. The RMSE is also calculated to assess model accuracy. I would be interested in seeing how this compares to some kind of baseline model (perhaps a simple seasonality / climatological model) although this would depend on the time units of the model presented here.

Resp: We addressed your suggestion in the discussion section: “The prediction capacity of our model gives a root-mean-square deviation of 33.9% per year (and 19.8%, dropping the outlier years), in the prediction accuracy of the total infection cases. There are some example of statistical models which depends of meteorological variables that produces more accurate predictions, like the work published by Ling Hii et al. [53], which reports an accuracy of 0.3% in a 16-weeks forecasting, or the model published by Naher et al. [54], which reports an accuracy fit of 10.8% in a 2-year time-series model. Nevertheless, our model predicts the proportions of DENV serotypes as well as the total amount of DENV infection over a 18-year time-lapse, with an accuracy allowing for long-term public health decisions.”

7. It’s still unclear to me the way in which the different serotypes are differentially affected by temperature as mentioned in line 267, some more discussion on this would be helpful.

Resp: We apologize for leaving the statement about the differential serotype outcome in the last version of the manuscript. We deleted this affirmation in this corrected version of the article.

Minor comments:

- The use of the term ‘dispersal’ in the title is confusing as it suggests a spatial analysis, I would recommend different terminology used by authors later on in the text such as dynamics or prevalence.the

Resp: We took your suggestion into account, we changed the title of the article to “Mathematical modeling of Dengue virus serotypes propagation in Mexico”

- In line 5 I think rather than ‘constitute’ the authors mean ‘constitutes’.

Resp: We corrected the point mentioned by the revisors.

- In line 33 authors refer to the dengue vaccine displaying ‘immunity’ but I believe ‘protection’ or ‘efficacy’ would be less confusing.

Resp: We corrected the point mentioned by the revisors.

- In line 164 “were” should be replaced with “where”

Resp: We corrected the point mentioned by the revisors.

- In line 184 the sentence “we graphed the temperature” could be replaced with “we displayed the temperature”

Resp: We corrected the point mentioned by the revisors.

- In line 274 the term “man made action” is unclear and could be replaced by “public health interventions”.

Resp: We corrected the point mentioned by the revisors.

---

## [Editor Report · Decision Letter 2]

26 Jun 2023

Mathematical modeling of Dengue virus serotypes propagation in Mexico

PONE-D-22-28107R2

Dear Dr. Condé,

We’re pleased to inform you that your manuscript has been judged scientifically suitable for publication and will be formally accepted for publication once it meets all outstanding technical requirements.

Kind regards,

Jan Rychtář

Academic Editor

PLOS ONE

Additional Editor Comments (optional):

all reviewers' comments were adequately addressed
---

## [Editor Report · Acceptance letter]

5 Jul 2023

PONE-D-22-28107R2 

Mathematical modeling of Dengue virus serotypes propagation in Mexico 

Dear Dr. Condé:

I'm pleased to inform you that your manuscript has been deemed suitable for publication in PLOS ONE. Congratulations! Your manuscript is now with our production department. 

Kind regards, 

on behalf of

Dr. Jan Rychtář 

Academic Editor

PLOS ONE